# CD112 Regulates Angiogenesis and T Cell Entry into the Spleen

**DOI:** 10.3390/cells10010169

**Published:** 2021-01-15

**Authors:** Erica Russo, Peter Runge, Neda Haghayegh Jahromi, Heidi Naboth, Angela Landtwing, Riccardo Montecchi, Noémie Leicht, Morgan Campbell Hunter, Yoshimi Takai, Cornelia Halin

**Affiliations:** 1Institute of Pharmaceutical Sciences, ETH Zurich, Vladimir-Prelog-Weg 1-5/10, CH-8093 Zurich, Switzerland; erica.russo@pharma.ethz.ch (E.R.); peter.runge@pharma.ethz.ch (P.R.); neda.haghayegh@pharma.ethz.ch (N.H.J.); heidi.engel89@gmail.com (H.N.); angela.landtwing@gmail.com (A.L.); riccardo.montecchi@pharma.ethz.ch (R.M.); leichtnoemie@gmail.com (N.L.); morgan.hunter@pharma.ethz.ch (M.C.H.); 2Department of Biochemistry and Molecular Biology, Kobe University Graduate School of Medicine, Chuo-ku, Kobe 650-0047, Japan; ytakai@med.kobe-u.ac.jp

**Keywords:** nectin-2, blood vessels, angiogenesis, spleen, T cell homing

## Abstract

Junctional adhesion proteins play important roles in controlling angiogenesis, vascular permeability and leukocyte trafficking. CD112 (nectin-2) belongs to the immunoglobulin superfamily and was shown to engage in homophilic and heterophilic interactions with a variety of binding partners expressed on endothelial cells and on leukocytes. Recent in vitro studies suggested that CD112 regulates human endothelial cell migration and proliferation as well as transendothelial migration of leukocytes. However, so far, the role of CD112 in endothelial cell biology and in leukocyte trafficking has not been elucidated in vivo. We found CD112 to be expressed by lymphatic and blood endothelial cells in different murine tissues. In CD112-deficient mice, the blood vessel coverage in the retina and spleen was significantly enhanced. In functional in vitro studies, a blockade of CD112 modulated endothelial cell migration and significantly enhanced endothelial tube formation. An antibody-based blockade of CD112 also significantly reduced T cell transmigration across endothelial monolayers in vitro. Moreover, T cell homing to the spleen was significantly reduced in CD112-deficient mice. Overall, our results identify CD112 as a regulator of angiogenic processes in vivo and demonstrate a novel role for CD112 in T cell entry into the spleen.

## 1. Introduction

In multicellular organisms, cell-to-cell contacts are essential for the development, plasticity and maintenance of tissues. Such cell-to-cell interactions are dynamically mediated through cellular junctions consisting of tight and adherens junctional proteins [1,2,3] and desmosomes [4]. Among other cell types, junctional connections are of particular importance in endothelial cells, which make up blood and lymphatic vessels. Endothelial cells are connected by adhesion molecules that ensure overall vessel integrity and control vascular permeability as well as leukocyte extravasation [5,6]. Besides cadherins, nectins constitute another emerging family of adhesion molecules localising to cell–cell junctions. Nectins belong to the immunoglobulin superfamily and can engage with a plethora of binding partners in either homophilic or heterophilic interactions on the same or opposing cells [7]. The four described nectin family members (nectin 1–4; also known as CD111-CD113 and PRR4) have been implicated in diverse cellular processes including proliferation, migration and adhesion, all of which are central to proper vascular function [8,9]. In endothelial cell biology, CD112 (also known as nectin-2) is of particular interest. Previous studies reported the expression of CD112 in vitro at the site of cellular junctions of human umbilical vein endothelial cells (HUVECs) [10]. CD112 expression was later confirmed in vivo in high endothelial venules (HEVs) in human lymph nodes (LNs) and in blood vessels of human skin [11]. CD112 interacts in a transhomophilic fashion with other CD112 molecules or in a transheterophilic fashion with CD113 (also known as nectin-3) expressed on adjacent cells [12]. Silencing of CD112 in human outgrowth endothelial cells in vitro led to increased cell proliferation, suggesting a regulatory role for CD112 in angiogenesis [13]. Besides interactions with members of the nectin family, CD112 also binds other types of surface proteins expressed on immune cells. For example, different subsets of T cells reportedly express the CD112 binding partners CD226 (DNAM-1, DNAX accessory molecule-1), T cell immunoreceptor Ig tyrosine (TIGIT)-based inhibition motif (ITIM) domain and CD112R [14,15,16], which all mediate immune-activating or -inhibitory signalling in leukocytes. Evidence suggests that expression of CD112 and of other nectin-related family members (i.e., CD155) by endothelial cells regulates transendothelial migration of leukocytes. Specifically, human T cell and monocyte transmigration was shown to depend on CD112 and CD155, respectively [10,11]. However, thus far, these findings are limited to in vitro observations, as the in vivo role of CD112 in endothelial cell biology and in leukocyte trafficking has not been investigated.

In this work, we set out to further address the function of CD112 in (lymph)angiogenesis as well as in leukocyte trafficking. Performing functional in vitro assays and by analysing the blood and lymphatic vasculatures in CD112-deficient mice, we show that CD112 regulates angiogenic processes in vitro and in vivo. Furthermore, we report that CD112 serves as an adhesion molecule for murine T cell transmigration and homing to the spleen.

## 2. Materials and Methods

### 2.1. Mouse Strains

Wild type (WT) C57BL/6 mice were purchased from Janvier (Genest-Saint-Isle, France). CD112^−/−^ mice [17], littermate controls (WT) and hCD2-DsRed [18] mice were bred and housed in an optimal-hygiene (OHB) environment. All animal experiments were approved and performed according to the regulations of the Cantonal Veterinary Office Zurich (protocol numbers 268/2014 and 238/2017).

### 2.2. Cell Culture

All experiments with cells were conducted in sterile environments under the laminar airflow cabinet (HERA safe, Thermo Fisher Scientific, Waltham, MA, USA). Conditionally immortalized lymphatic endothelial cells (imLECs), expressing a heat-labile version of the large T antigen, were isolated from the back skin of H-2Kb-tsA58 (immorto) mice [19] as previously described [20]. imLECs were cultured at 33 °C in humidified air and 5% CO_2_ in 10 µg/mL collagen (PureCol, Advanced Biomatrix, San Diego, CA, USA) and 10 µg/mL fibronectin (Millipore, Temecula, CA, USA) precoated culture dishes (TPP, Trasadingen, Switzerland) in media supplemented with 40% Dulbecco’s modified eagle medium (DMEM; low glucose), 40% F12-Ham, 20% fetal bovine serum (FBS) (all from Gibco, Paisley, UK), 56 µg/mL heparin (Sigma-Aldrich, St Luis, MO, USA), 10 µg/mL endothelial cell mitogen (AbD Serotec, Duesseldorf, Germany), 1% antibiotic antimycotic solution (Fluka, Buchs, Switzerland), and L-glutamin (2 nM; Fluka). Murine IFN-γ 1U/mL (Peprotech, London, UK) was added to induce large T-antigen expression [19]. Forty-eight hours prior to experiments, imLECs were transferred to 37 °C and cultured in complete medium without IFN-γ, to induce degradation of the large T antigen [20].

Simian virus 40 (SV40)-transformed blood endothelial cells (BECs) isolated from pancreas islet of Langerhans (MS-1 cells [21]) were purchased from ATTC (Manassas, VA, USA). MS-1 were cultured at 37 °C in humidified air and 5% CO_2_ in dishes coated with 10 ug/mL of collagen (Advanced Biomatrix) and fibronectin (Millipore) in DMEM media (Gibco) supplemented with 5% FBS (Sigma-Aldrich) and 1% antibiotic-antimycotic solution (Fluka).

Human primary lymphatic endothelial cells (LECs) were kindly provided by Dr. Michael Detmar, ETH Zurich [22]. Cells were cultured on plates coated with 10 µg/mL collagen type I (PureCol, Advanced BioMatrix) and fibronectin (Millipore) in EBM (Lonza, Basel, Switzerland) supplemented with 20% foetal bovine serum (Gibco), antibiotic-antimycotic solution (Fluka), L-glutamine (2 mM; Fluka), hydrocortisone (10 mg/mL; Fluka), and N,6 29-O-dibutyryladenosine 39,59-cyclic monophosphate sodium salt (cAMP, 2.5 × 10^−2^ mg/mL; Fluka). All cells were maintained at 37 °C in humidified air with 5% CO_2_.

### 2.3. Isolation and Cultivation of Primary LN Endothelial Cells

LN endothelial cells were isolated as previously described [23]. In brief, skin draining LNs (popliteal, inguinal, axillary, brachial and auricular) were collected from WT and CD112^−/−^ mice. LNs were digested in Roswell Park Memorial Institute (RPMI) medium (Gibco) supplemented with 0.25 mg/mL Liberase DH (Roche, Basel, Switzerland) and 200 U/mL DNase I (Sigma-Aldrich) for 1 h at 37 °C and 5% CO_2_. During incubations, cell suspensions were mixed every 15 min by pipetting up and down. After digestion, LN cell suspensions were passed through a 70 µm cell strainer (Invitrogen) and plated into cell culture dishes precoated with 10 µg/mL collagen type I (PureCol, Advanced BioMatrix) and 10 µg/mL fibronectin (Millipore) in Minimal Essential Medium (MEM)-alpha medium containing 10% FBS and 1× penicillin/streptomycin (all from Gibco). Cells were cultured at 37 °C and 5% CO_2_ until confluency and the medium was changed 24 and 72 h after digestion. At confluency, cells were collected with Accutase (Sigma-Aldrich) and endothelial cells were isolated using CD31^+^ microbeads (Miltenyi Biotech, Bergisch Gladbach, Germany). Purity was determined with flow cytometry and endothelial cells were propagated up to passage 6 after isolation.

### 2.4. Flow Cytometry

In vitro cultured MS-1, imLECs and primary LN LECs were washed with phosphate-buffered saline (PBS) and detached with Accutase (Sigma-Aldrich). Unspecific FcRγ binding was blocked with anti-mouse CD16/32 antibody (Biolegend, Luzern, Switzerland), prior to staining with the following primary antibodies and corresponding isotype controls: 20 µg/mL mouse anti-human CD112 (clone: R2.525, Santa Cruz Biotechnology, Heidelberg, Germany) (isotype: Mouse IgG1; clone MG1-45, Biolegend), 2.5 µg/mL rat anti-mouse CD112-Alexa Fluor 488 (clone: 829038, R&D systems, Abingdon, UK) (isotype: Alexa Fluor Rat IgG2a; clone RTK2758, Biolegend). Alexa Fluor 488-conjugated secondary antibodies (Caltag Laboratories, Little Balmer, UK) were used for detection in case of uncoupled primary antibodies. All incubations were performed on ice in fluorescent activated cell sorting (FACS) buffer containing 2% FBS and 2 mM ethylenediaminetetraacetic acid (EDTA) diluted PBS.

Single-cell suspensions of mouse tissues (ears, LNs and spleens) were obtained by enzymatic digestion with 4 mg/mL collagenase IV (Invitrogen, Basel, Switzerland) for 45 min at 37 °C and subsequently passed through a 40 µm cell strainer (Invitrogen), as described previously [20]. Cell suspensions were stained with rat anti-mouse CD31-APC, rat anti-mouse CD45-PerCP, hamster anti-mouse podoplanin-PE/Cy7, rat anti-mouse mucosal vascular addressin cell-adhesion molecule 1 (MadCAM-1)-PE, rat anti-mouse PNAd-Biotin plus Streptavidin APC/Cy7, rat anti-mouse ICAM-1-FITC (all from BioLegend) and rat anti-mouse CD112-Alexa Fluor 488 (R&D system) or corresponding isotype controls.

To quantify MadCAM-1^+^CD31^+^ BECs, spleens were digested with 4 mg/mL collagenase IV (Invitrogen) for 45 min at 37 °C and subsequently passed through a 40 µm cell strainer (Invitrogen). Single-cell suspensions were stained with rat anti-mouse CD31-APC, rat anti-mouse CD45-PerCP, hamster anti-mouse podoplanin-PE/Cy7, and rat anti-mouse MadCAM-1-PE. The cell numbers measured in FACS were normalized to the total spleen cell number manually counted with a Neubauer counting chamber (depth: 0.100 mm, area: 0.0025 mm^2^). For CD112 binding partner analysis, murine CD4^+^ T cells were isolated from spleen single-cell suspensions using CD4 (L3T4) microbeads (Miltenyi Biotec) after red blood cell lysis with Ammonium-Chloride-Potassium (ACK) buffer (150 mM NH_4_Cl, 10 mM KHCO_3_, 0.1 mM Na_2_EDTA, pH 7.4). CD4^+^ T_H_1 cells were generated as previously described [24]. Briefly, purified CD4^+^ T cells were cultured on plate-bound hamster anti-mouse CD3 (3 µg/mL, 145-2C11) and hamster anti-mouse CD28 (5 µg/mL, 37.51) (both from BioLegend) with murine IFN-γ (20 ng/mL, Peprotech), interleukin (IL)-12 (5 ng/mL, Peprotech) and rat anti-mouse IL4 (5 µg/mL, 11B11, BioLegend) for 3 days and then transferred to a new 6-well plate with fresh media and cultured for additional 2 days at 37 °C. Prior to staining, dead cells were removed with Ficoll (Ficoll paque plus, Sigma-Aldrich) density centrifugation. T cells were stained with rat anti-mouse CD45-APC/Cy7 (30-F11), rat anti-mouse CD4-Alexa Fluor 700 (GK1.5), rat anti-mouse CD44-FITC (IM7), rat anti-mouse CD62L-APC (MEL-14), rat anti-mouse CD226-PE/Cy7 (10E5) and mouse anti-mouse TIGIT-BV421 (1G9) (all from BioLegend) and corresponding isotype controls.

Flow cytometry data were acquired on a BD FACS Canto (BD Biosciences, New Jersey, USA) using FACS Diva software (BD Biosciences) or a CytoFLEX S Flow Cytometer (Beckman Coulter, Brea, CA, USA). Data were analysed offline using FlowJo software (Treestar, Ashland, TN, USA). In some experiments, delta median fluorescent intensity (MFI) values, defined as the difference in the MFI between the candidate and the corresponding isotype control staining, were calculated.

### 2.5. Immunofluorescence Staining of imLECs

imLECs were grown to 80–90% confluency in chamber slides (Nunc International, Naperville, IL, USA) coated with 10 µg/mL collagen and fibronectin as described before. Cells were incubated for 24 h in starvation medium (2% FCS), and thereafter the monolayers were washed with PBS, fixed with 4% paraformaldehyde (PFA) and permeabilized with 0.1% Triton-X100-PBS solution. Cells were incubated overnight with goat anti-mouse VE-cadherin and Alexa-488-conjugated rat anti-mouse Nectin-2 (R&D system), followed by detection with secondary antibodies (Invitrogen). Cell nuclei were stained with PBS/Hoechst 33342 trihydrochloride (Invitrogen). Monolayers were washed and mounted using Vectashield mounting medium (Vector Laboratories, Burlingame, MA, USA) and analysed on a Leica TCS SP8 confocal microscope (Leica Microsystems) using a 63× objectives (Leica, NA 1.4).

### 2.6. Western Blot

Human LECs (Promocell) were cultured on collagen/fibronectin precoated plates in EGM-2 medium (Lonza) supplemented with FBS, bFGF, EGF, IGF-1, hydrocortisone, ascorbic acid, gentamicin, and amphotericin-B (all from Lonza: EGM-2 supplements). At confluency, cells were washed twice with PBS and lysed in lysis buffer (RIPA, Thermo Fisher) containing complete Mini EDTA-free Protease Inhibitor Cocktail (Sigma-Aldrich) for 10 min on ice. Cell lysates were harvested and incubated for 15 min on ice, centrifuged and supernatant was collected. Samples were diluted in reducing Lämmli buffer (1:4, prediluted 1:10 with Beta-Mercaptoethanol, BioRad, CA, USA) and cooked for 5 min at 95 °C. Subsequently, samples were run on an SDS-PAGE precast gel (4–20% acrylamide, BioRad) at 80 V for 1 h and 45 min. Gels were transferred onto nitrocellulose membranes at 19 V for 20 min. To block unspecific binding sites, membranes were incubated in 5% milk/PBS overnight. Mouse anti-human CD112 (E-1, Santa Cruz) was used in 1:100 and anti-β-actin (Abcam) at 1:5000 for 2 h at room temperature. Secondary anti-mouse and anti-rabbit horseradish peroxidase (HRP) conjugated secondary antibodies (both Thermo Fisher) were used at 1:1000 for 1 h at room temperature. Signals were detected using Clarity Western ECL substrate (BioRad) and blots were developed with a ChemiDoc MP Imaging System (BioRad). Blots were cut at 50kDa mark to allow simultaneous staining for CD112 and β-actin. Image analysis was performed using the Image Lab software (BioRad).

### 2.7. Quantitative Real-Time PCR

Human LECs (Promocell) were cultured as described above. RNA was isolated using Trizol according to the manufacturer’s protocol (Life technologies, Carlsbad, MA, USA). To extract total tissue RNA from spleens of WT and CD112^−/−^ mice, spleens were snap frozen on liquid nitrogen and homogenized in Trizol twice for 30 s at maximum speed using a Tissue Lyser (Qiagen, Hombrechtikon, Switzerland). Cell lysates were then centrifuged and supernatant was collected for RNA isolation according to the Trizol manufacturer’s protocol (Life technologies). Isolated RNA was reverse transcribed using the High Capacity cDNA kit (Thermo Fisher). Quantitative real-time PCR was performed in triplicates on a Fast Real-Time-PCR System (Applied Biosystems, Thermo Fisher) using SYBR Green (Roche). Human β-actin (Thermo Fisher) was used as an internal control. Primer sequences: hCD112 fw: CGGAACTGTCACTGTCACCA, rev: GACACTTCAGGAGGGTAGCG; mouse VEGF-A fw: CGTACTTGCAGATGTGACAAGCCAA, rev: CGGTGACGATGATGGCGTGGT; mouse Angiopoetin-1 fw: TGCAGCAACCAGCGCCGAAA, rev: CAGGGCAGTTCCCGTCGTGT; mouse Angiopoetin-2 fw: GCTTCGGGAGCCCTCTGGGA, rev: CAGCGAATGCGCCTCGTTGC.

### 2.8. Scratch Assay

Human LECs were grown until confluency in collagen/fibronectin coated 24-well plates and then incubated for 24 h in starvation medium (EBM, supplemented with 2% FCS and 1% antibiotic-antimycotic solution). To perform the assay, a cross-shaped “scratch” was applied to each monolayer using a sterile 200 μL pipette tip. Monolayers were washed twice with PBS and 500 μL of starvation medium supplemented with 20–50 ng/mL VEGF-A (Peprotech) and/or mouse anti-human CD112 (R2.525, Santa Cruz) were added (4 replicates per condition). Representative images of the cell-free zones were acquired immediately after the scratch application 18–22 h later. For the analysis, the percentage of the surface area closed at the indicated time points (between 18–22 h) was quantified using T-scratch software (Geback et al., 2009).

### 2.9. Tube Formation Assay

Human LECs were cultured in a collagen/fibronectin precoated 24-well plates as described above. At confluency, monolayers were washed with PBS and incubated in starvation EBM-2 medium (Lonza) comprising 5% FBS, 1% antibiotics and 1% L-glutamine (200 mM) (all from Gibco) for 6–10 h at 37 °C in humidified air and 5% CO_2_. For tube formation induction, cell monolayers were overlaid with 0.5 mL of isotonic extracellular matrix solution (1 mg/mL collagen diluted in EBM-2 starvation medium) containing either 5 µg/mL mouse anti-human CD112 blocking antibody (R2.525, Santa Cruz) or the corresponding isotype control (experiment was performed in triplicates per condition). Representative images (3 per well) were acquired after 16–18 h on a fluorescent microscope (Nikon Eclipse Ti-E, Tokio, Japan) using a 4× objective and 1.5 zoom and the total length of tubes in the field of view which was quantified manually using a self-made macro compatible for FIJI (ImageJ).

### 2.10. Permeability Assay

Human LECs were grown to confluency on transwell membrane inserts with pore sizes of 0.4 μm (Transwell Permeable Support, polycarbonate membrane, 6.5 mm diameter inserts, Costar). Twenty-four hours prior to the experiment, cells were incubated with starvation EBM-2 medium (5% FCS, 1% antibiotics and 1% L-glutamine). Afterwards, monolayers were pretreated for 1 h with either 5 μg/mL of mouse anti-human CD112 (R2.525, Santa Cruz) or 20 ng/mL of recombinant human IFNγ (PeproTech, 100 μg/mL) (positive control) diluted in 100 μL starvation medium and bottom wells were filled with 650 µL starvation medium. Permeability assay was induced by adding 50 μL of 2.5 mg/mL FITC-dextran (Sigma-Aldrich, 70 kDA, FD70S) to the upper transwell inserts and cells were incubated for 15 and 30 min at 37 °C. Afterwards, 75 µL was taken from the bottom chamber and the fluorescence absorbance of FITC-dextran was measured separately for both readouts at 490 nm excitation wavelength and 520 nm emission wavelength on a SpectraMax Gemini EM (Bucher Biotech AG, Basel, Switzerland). Raw data were processed and presented as fold change compared to the isotype control.

### 2.11. siRNA Transfection Protocol

In total, 40,000 human LECs (Promocell) were seeded in a 24-well plate and cultured as described above. At 80% confluency, cells were washed with PBS and incubated with Opti-MEM medium (Thermo Fisher) supplemented with either 10 nM Silencer Select Negative Control #1 small inhibitory RNA (siRNA) (Cat. 4390843), 10 nM Silencer Select human CD112 siRNA #1 (ID s11606) or 10 nM Silencer Select human CD112 siRNA #2 (ID s11607) for 6 h at 37 °C. Previous to transfection, siRNAs were incubated with Lipofectamine (RNAiMax, Thermo Fisher) for 15 min at room temperature. After transfection, cells were kept in culture medium overnight. Transfection efficiency was determined by flow cytometry. Cells were detached with Accutase (Sigma-Aldrich) and incubated with mouse anti-human CD112 (20 µg/mL, R2.525, Santa Cruz) and the respective isotype control (R&D system). Alexa Fluor 488-conjugated secondary antibody was used for detection. All incubations were performed on ice in FACS buffer containing 2% FBS and 2 mM EDTA diluted PBS. Samples were acquired on a CytoFLEX S Flow Cytometer (Beckman Coulter). Data were analysed offline using FlowJo software (Treestar).

### 2.12. Whole Mount Immunofluorescence of Ear and Retina

Mice were euthanized, and ears were depilated using Veet cream and harvested. Ears were split into 2 halves along the cartilage and fixed at room temperature for 2 h in 4% PFA. Ear halves were subsequently washed with 0.3% triton-X/PBS and blocked in immunomix (IM) containing 0.1% bovine serum albumin (BSA)/PBS and 5% normal donkey serum (Sigma-Aldrich). Ears were subsequently incubated overnight at 4 °C in IM with either Rat anti-mouse CD112 (R&D systems), goat anti-mouse VE-cadherin (R&D), rabbit anti-mouse LYVE-1 (Angiobio, San Diego, CA, USA) or with corresponding isotype controls.

For immunofluorescent staining of retinal blood vessels, postnatal day 6 (P6) retina tissues were harvested, fixed and blocked as described above. Blood vessels were stained overnight at 4 °C with biotinylated isolectin B4 (iB4, Vector Laboratories) diluted in PBS. Afterwards, retinas were incubated for 3 h with Alexa Fluor 488-conjugate antibody (Invitrogen) or Alexa Fluor 488-conjugated streptavidin (Invitrogen) in presence of Hoechst 33,342 (Invitrogen). Samples were washed with PBS and mounted using Mowiol (Vector Laboratories). Images were acquired on a Leica TCS SP8 confocal microscope (Leica Microsystems) 40× and 63× objectives (NA 1.4) or on a Nikon Eclipse Ti-E (Tokyo, Japan) equipped with a Hamamatsu ORCA-Flash4.0 CCD camera (Hamamatsu, Japan) and a 4× objective. Images were acquired using LAS AF version 4.0 software (Leica Microsystems) and analysed with Imaris software (version 7.1.1; Bitplane, Zurich, Switzerland). Analysis of blood vessel coverage and length within the retina was performed manually in a blinded manner using Fiji (ImageJ, USA).

For morphometric analysis of blood vessels in the ear skin, blood vessels were stained overnight at 4 °C with 3 µg/mL rat anti-mouse MECA-32 in PBS (BD Bioscience). The following day, the tissues were incubated for 3 h with donkey anti-rat Alexa Fluor 488-conjugated secondary antibodies (Invitrogen). Samples were washed with PBS and mounted using Mowiol (Vector Laboratories). Images were acquired on a Leica TCS SP8 confocal microscope using a 10× objective with 1.5 zoom (Leica Microsystems). Computer-assisted image analysis of ear skin was performed with AutoTube software [25].

### 2.13. Spleen Immunofluorescence

Spleens from WT and CD112^−/−^ mice were harvested and embedded in optimum cutting temperature (OCT) compound (Tissue-TEK, Sakura Finetek, Torrance, CA, USA) and frozen on liquid nitrogen. Spleens were cut in half and re-embedded in OCT. Then, 50 µm sections were prepared on a CryoStar NX50 (Thermo Fisher) (4 sections per spleen) and fixed for 2 min in Aceton at −20 °C and subsequently for 5 min in MeOH at 4 °C. Afterwards, sections were washed twice with PBS for 10 min and unspecific binding was blocked with 2% BSA (Sigma-Aldrich) supplemented with 5% normal donkey serum (Sigma-Aldrich) and 0.1% Tween-20 (Sigma-Aldrich) followed by incubation in primary antibodies diluted in blocking solution overnight at 4 °C: rat anti-mouse MadCAM-1-PE (MECA-367) (2.5 µg/mL) and rat anti-mouse pan-endothelial cell antigen (MECA-32) (3 µg/mL) (Biolegend and BD Bioscience). The next day, sections were washed 3× with PBS and incubated with Alexa Fluor 488-conjugated secondary antibodies (Invitrogen) diluted in PBS for 1 h at room temperature. After washing 3× with PBS, sections were mounted in Vectashield (Vector Laboratories). In total, 3 images/section from 3–4 sections/mouse were acquired on a fluorescent microscope (Nikon Eclipse Ti-E, Tokio, Japan) using a 10× objective and 1.5 zoom. Analysis of blood vessel coverage was performed manually using Fiji (ImageJ, USA).

### 2.14. Transendothelial Migration Assay

Endothelial cells (MS-1, human LECs or primary LN LECs) were grown to confluency on collagen/fibronectin (10 µg/mL) precoated transwell membrane inserts with a pore size of 5 µm (Corning Life Sciences, NY, USA). Before starting the assay, endothelial cell monolayers were pretreated with blocking antibodies or the corresponding isotype controls for 30 min at 37 °C: CD112 was blocked with 20 µg/mL mouse anti-human CD112 (R2.525, Santa Cruz) and ICAM-1 was blocked serving as a positive control with 10 µg/mL rat anti-mouse CD54 (YN1/1.7.4, Biolegend). Each condition was performed in triplicates. In the case of human LECs, 10 µg/mL mouse anti-human CD54 (BBIG-I1 (11C81), R&D Systems) was used as a positive control. Designated isotype controls for mouse anti-human CD112 (Mouse IgG1; clone MG1-45, Biolegend or normal mouse IgG1, Santa Cruz), mouse anti-human CD54 (Mouse IgG1; clone MG1-45, Biolegend) and anti-mouse CD54 (Rat IgG2b, k; clone RTK4530, Biolegend or clone eB149/10H5, eBioscience) were used in the control condition. In total, 50,000 T cells were added to the top wells in presence of blocking antibodies. For transmigration through MS-1 monolayers, murine CD4^+^ T cells were freshly isolated from LNs and spleen using CD4 (L3T4) microbeads (Miltenyi Biotec). For transmigration through human LECs, human peripheral blood mononuclear cells (PBMCs) were isolated from buffy coats (purchased from the Blood Donation Center Zurich, Switzerland) using Ficoll (Ficoll paque plus, Sigma-Aldrich) density gradient centrifugation. Transmigration was induced by adding 100 ng/mL CCL21 in EBM complete medium to the bottom wells. After 4 h, transmigration was stopped and the medium was collected in the bottom well. Cells were stained with rat anti-mouse CD4-APC antibody for 20 min at 4 °C and quantified by flow cytometry on a BD FACS Canto (BD Biosciences) or CytoFLEX S Flow Cytometer (Beckman Coulter).

### 2.15. T Cell Homing Experiment

Single-cell leukocyte suspensions were prepared from spleen and peripheral LNs of hCD2-dsRED mice [18]. T cell homing was started by injecting 1 × 10^7^ leukocytes intravenously (iv) into WT and CD112^−/−^ mice. After 2.5 h, mice were sacrificed and secondary lymphoid organs (SLOs) such as peripheral LNs (PLNs: 2 brachial, 2 auricular and 2 inguinal LNs pooled), mesenteric LNs (MLNs, *n* = 3), Peyer’s Patches (PP, *n* = 3) and the spleen were collected. Single-cell suspensions were obtained by smashing the SLOs through a 40 µm cell strainer (Alibaba Group, Hangzhou, China) using a syringe plunger. Subsequently, cells were stained with the following rat anti-mouse monoclonal antibodies (all from Biolegend): anti-CD45-PerCP (30-F11), anti-CD4-APC (GK1.5) and anti-CD8α-FITC (53-6.7). The number of adoptively transferred CD2-dsRED^+^ T cells was quantified on a BD FACS Canto (BD Biosciences) or CytoFLEX S Flow Cytometer (Beckman Coulter). Total tissue cell number was assessed manually with a Neubauer’s counting chamber. CD2-dsRED^+^ T cell numbers homed into the SLOs were normalized to the manual cell counts.

### 2.16. Spleen Section Analysis

For the microscopic analysis of T cell homing into the spleen of WT and CD112^−/−^ mice, CD4^+^ T cells were purified from the LNs and spleens of C57Bl/6 mice using CD4 (L3T4) microbeads (Miltenyi Biotec). CD4^+^ T cells were labelled with 5 µM cell proliferation dye eFluor 670 (eBioscience, San Diego, CA, USA) for 25 min at 37 °C in plain RPMI medium (Thermo Fisher). After extensive washes, 0.5–1 × 10^6^ cells were injected intravenously into CD112^−/−^ and WT mice. After 2.5 h, spleens were harvested and embedded in an optimum cutting temperature (OCT) compound (Tissue-TEK, Sakura Finetek) and frozen on liquid nitrogen as described above. As control, PLNs were harvested and T cell homing was analysed by flow cytometry, the same as previously explained. Spleens were cut in half and re-embedded in the OCT compound. Sections 50 µm in size were prepared on a CryoStar NX50 (Thermo Fisher) (3–4 sections per spleen) and fixed for 2 min in acetone at −20 °C and subsequently for 5 min in MeOH at 4 °C. Afterwards, sections were washed twice with PBS for 10 min and unspecific binding was blocked with 2% BSA (Sigma-Aldrich) supplemented with 5% normal donkey serum (Sigma-Aldrich) and 0.1% Tween-20 (Sigma-Aldrich) followed by incubation in primary antibodies diluted in blocking solution overnight at 4 °C: anti-B220-Alexa Fluor 647 (RA3-6B2), anti-CD4-biotin (RM4–5) (both Biolegend). The next day, sections were washed 3× with PBS and incubated with secondary antibodies diluted in PBS: Streptavidin-Alexa Fluor 594 (Biolegend) for 1 h at room temperature. After washing 3× with PBS, sections were mounted in Vectashield (Vector Laboratories). In total, 6–10 images/section from 3–4 sections/mouse were acquired on a confocal microscope (Zeiss LSM 780) using a 20× objective (0.8 NA Plan-Apochromat M27) and 0.6 zoom. The number of homed T cells into the spleen per image was analysed in a blinded manner using the particle analyser in Fiji and was normalized on the T cell zone area.

### 2.17. Statistical Analysis

Graphs were generated and statistical analysis was performed with Prism 7 (GraphPad, San Diego, CA, USA). Data sets were analysed using the student t-test (unpaired, two-tailed) when comparing two groups and one-way ANOVA when comparing three or more groups. Unless stated otherwise, the data are shown as mean and standard error of mean (SEM). Differences were considered statistically significant when *p* ≤ 0.05.

## 3. Results

### 3.1. CD112 Is Expressed by Blood and Lymphatic Vasculature in Murine Tissues

A transcriptomics study performed by our lab [20] suggested that CD112 expression in blood vascular and lymphatic endothelial cells (BECs and LECs, respectively) derived from murine skin. To further investigate this finding, we performed flow cytometry analysis of single-cell suspensions generated from murine ear skin. This analysis confirmed CD112 protein expression in both LECs and BECs (Figure 1A,B), with consistently higher expression levels in LECs (Figure 1C). Flow cytometry analyses also detected CD112 expression in endothelial cells present in murine LNs and spleen. By contrast, no signal was found when staining endothelial cells in single-cell suspensions generated from tissues of CD112-deficient mice, demonstrating the specificity of our anti-CD112 antibody (Appendix A). CD112 expression in both blood and lymphatic vessels was also confirmed by whole mount immunofluorescence performed on murine ear skin and diaphragm (Figure 1D, Appendix A). In lymphatic and blood vessels, CD112 expression followed the distinct junctional organisation previously described for the endothelial cell-specific marker VE-cadherin [1] (Figure 1D). Particularly at the level of lymphatic capillaries, which are characterized by their expression of the lymphatic vascular endothelial hyaluronan receptor–1 (LYVE-1), CD112 protein was present at the site of the discontinuous “button”-like junctions, whereas in elongated LYVE-1 negative lymphatic collector vessels, CD112 was expressed at the site of the continuous “zipper”-like junctions (Figure 1E).

Flow cytometry analysis of in vitro cultured human and murine LECs further confirmed the expression of CD112 at the protein level. Interestingly, CD112 cell surface expression was upregulated upon stimulation with the inflammatory mediators IFNγ and TNFα, as evidenced by flow cytometry (Figure 2A,B). In contrast, no upregulation of total CD112 protein upon IFNγ and TNFα treatment was detected by Western blot (Figure 2C,D). Similarly, in quantitative real-time PCR analysis, CD112 mRNA levels remained unaltered upon TNFα and IFNγ treatment of human LECs (Figure 2E). When performing immunofluorescence on in vitro cultured immortalized murine LECs, we found CD112 to localize to sites of cell–cell contacts, again displaying a similar expression pattern as the adherens junction molecule VE-cadherin (Figure 2F). Since CD112 binds in trans to CD113, which reportedly is expressed on HUVECs in vitro [11], we also investigated CD113 expression on endothelial cells in vitro and in vivo. While we detected CD113 expression in in vitro cultured human and murine LECs (Appendix A), flow cytometry analysis and whole mount immunofluorescence performed on ear skin did not reveal any CD113 protein on lymphatic and/or blood vessels in vivo (Appendix A), in agreement with our transcriptomics data [20]. Overall, our findings demonstrated the expression of the junction molecule CD112 in BECs and LECs in various murine tissues.

### 3.2. CD112 Regulates Endothelial Cell Migration and Permeability In Vitro

CD112 was previously found to impact proliferation, migration and tube formation of human outgrowth endothelial cells in vitro [13]. To further investigate the role of CD112 in endothelial cell biology, we performed various cell-based assays with skin-derived human LECs in the presence of an anti-CD112-blocking antibody (R2.525 [26]). Notably, the specificity of the R2.525 antibody for CD112 was confirmed by performing flow cytometry analysis on human LECs, in which CD112 had been knocked-down by siRNA (Appendix A). To assess the contribution of CD112 to LEC migration, a cell-free scratch was introduced into confluent LEC monolayers and the capacity of bordering cells to close the cell-free zone was quantified after 18–22 h (Figure 2G). Addition of the proangiogenic molecule VEGF-A greatly increased scratch closure, whereas treatment with anti-CD112 significantly reduced VEGF-A-induced wound closure (Figure 2H). Following up on this result, we investigated whether CD112 would also regulate the formation of endothelial tubes into a collagen matrix. In agreement with previous findings [13], tube formation was significantly enhanced upon antibody-mediated blockade of CD112 (Figure 2I). To investigate the role of CD112 in mediating endothelial cell barrier function, we next performed an in vitro permeability assay, by quantifying the diffusion of a fluorescent dye (FITC-dextran, 70 kDa) across confluent human LEC monolayers. Upon treatment with anti-CD112 antibody, endothelial permeability was significantly enhanced to a similar degree as seen upon treatment with IFN-γ (Figure 2J). Overall, our findings confirmed an involvement of CD112 in regulating endothelial cell processes in vitro.

### 3.3. Blood Vessel Coverage Is Increased in the Retina and Spleen of CD112^−/−^ Mice

Based on our in vitro findings, we investigated whether CD112 contributes to vascular network formation in vivo. Surprisingly, analysis of the lymphatic network in whole-mount samples prepared from ear skin of CD112^−/−^ and WT mice revealed no apparent changes in lymphatic vessel area, length or diameter (Appendix A). When investigating whether any compensatory mechanisms might be at play, we found that the expression of VE-cadherin was significantly increased in ear skin LECs of CD112^−/−^ mice (Appendix A). However, a subtle increase in the area covered by blood vessels and in the number of blood vascular intersections was observed in the ear skin of CD112^−/−^ mice (Appendix A). Since the formation of new vessels is particularly crucial during tissue development, we analysed the retinas of WT and CD112^−/−^ postnatal 6 (P6) pups. When staining the retinal blood vessel network with the endothelial cell-specific isolectin B4 (iB4) [27] (Figure 3A), we observed that CD112 deficiency significantly increased retinal blood vessel coverage (Figure 3B) and the overall length of the retinal plexus (Figure 3C). Moreover, the peri-arterial distance of the blood vessel network, namely the length of a blood vessel sprouting outwards from the artery, was significantly reduced in CD112^−/−^ mice (Figure 3D,E). Next, we analysed the spleens of adult CD112^−/−^ mice for a vascular phenotype. Notably, we did not detect any significant differences in the weight or cellularity of spleens from CD112-deficient as compared to WT mice (Appendix A). Immunofluorescence performed on spleen sections (Figure 3F,G) revealed a significantly enhanced blood vessel coverage in the absence of CD112 (Figure 3H,I). To confirm this result, we performed flow cytometry analysis of spleen single-cell suspensions and quantified the number of splenic BECs (Figure 3J,K). Both the numbers of podoplanin (podo)^−^CD31^+^ (Figure 3J) and of podo^−^CD31^+^MadCAM-1^+^ BECs (Figure 3K) were increased in CD112^−/−^ mice compared to WT littermates. To determine whether any compensatory upregulation of proangiogenic factors such as VEGF-A and angiopoietin (Ang)1 or Ang2 upon CD112-deficiency might account for this effect, we performed quantitative real time-PCR analysis of RNA isolated from whole spleen tissue. While VEGF-A expression was marginally decreased in CD112^−/−^ spleens, both Ang1 and Ang2 expressions were unaltered (Appendix A). In line with our previous findings (Appendix A), we observed that CD112 was expressed by all splenic BECs (Figure 3L,M). Interestingly, MadCAM-1^+^ BECs, which were reported to be one of the entry routes of lymphocytes into the splenic white pulp [28], also displayed high levels of CD112 (Figure 3M). Overall, we identified a role for CD112 in regulating the formation of blood vessels in mice during development and adulthood.

### 3.4. Blockade of CD112 Impairs CD4^+^ T Cell Transmigration through BECs In Vitro

Intrigued by the high expression of CD112 by splenic BECs (Figure 3M), we set out to further investigate the role of CD112 in leukocyte trafficking. Previous studies have already implicated CD112 in the transmigration of human T cells and monocytes across the blood vascular endothelium in vitro [10,11,26]. In line with these findings, antibody-mediated blockade of CD112 reduced human T cell transmigration through endothelial cell monolayers to a similar extent as seen upon blockade of ICAM-1 (positive control) (Figure 4A,B). Next, we investigated whether CD112 also regulated murine T cell migration across murine BECs. Flow cytometry analysis confirmed CD112 expression in the murine BEC cell line MS-1 [21] (Figure 4C). Unfortunately, no murine CD112-blocking antibody has been described to date. However, the high sequence homology between mouse and human CD112 [29] prompted us to test the binding and blocking ability of a previously described anti-human CD112 antibody (clone R2.525 [26]). Indeed, we found that the anti-human CD112 blocking antibody bound to CD112 expressing murine MS-1 cells (Figure 4D) and reduced CD4^+^ T cell transmigration across MS-1 cells in a comparable manner to anti-ICAM-1 blockade (positive control) (Figure 4E–H, Appendix A). Moreover, the transmigration of murine CD4^+^ T cell across primary CD112-deficient LECs isolated from the LNs of CD112^−/−^ mice was significantly reduced in comparison to transmigration across control LECs isolated from LNs of WT mice (Figure 4I,J).

In contrast to transmigration of freshly isolated CD4^+^ T cells (Figure 4E–J), transmigration of in vitro generated T_H_1 cells across MS-1 cells was not affected by the CD112 blockade (Appendix A). Based on these findings, we investigated the expression of the CD112 binding partners, CD112, CD113 and the immune receptors CD226, TIGIT and CD112R on freshly isolated as compared to in vitro activated murine T cells. While human lymphocytes reportedly express CD113, but not CD112 [11], surprisingly, neither CD112 nor CD113 could be detected on murine T cells (Appendix A). Flow cytometry analysis revealed that CD226 protein expression was considerably higher on freshly isolated splenic CD4^+^ T cells than on in vitro generated T_H_1 cells (Figure 5A), whereas TIGIT was not at all expressed on in vitro generated T_H_1 cells and only present in a minute fraction of CD4^+^ T cells freshly isolated from spleen (Figure 5B). We could also not detect CD112R by flow cytometry (data not shown). When analysing CD226 and TIGIT expression in greater detail in endogenous, spleen-derived CD4^+^ T cells, which comprise naïve (CD44^lo^CD62L^hi^), effector/memory (CD44^hi^CD62L^lo^) and central T cells (CD44^hi^CD62L^hi^) (Figure 5C), we found that all subsets expressed CD226, whereas TIGIT was only expressed on some effector/memory T cells (Figure 5D–F). Overall, these findings suggest that CD226 is the primary ligand for CD112 in regulating murine T cell transmigration across BECs.

Given the reduction observed in T cell transmigration in vitro, we next analysed the involvement of CD112 in T cell homing from blood to secondary lymphoid organs (SLOs) in mice. To this end, we intravenously injected lymphocytes isolated from hCD2-dsRED^+^ mice [18], which express dsRED protein in all T cells, into CD112^−/−^ and WT mice. 2.5 h after injection, we harvested the spleen, payer’s patches (PPs), mesenteric LNs (MLNs) and peripheral LNs (PLNs) and quantified the number of homed CD2-dsRED^+^ CD4^+^ and CD8^+^ T cells by flow cytometry (Figure 6A,B). T cell homing to the spleen, but not to other SLOs was significantly reduced in CD112^−/−^ mice compared to WT littermates (Figure 6C). This homing defect equally affected both CD4^+^ and CD8^+^ T cells (Figure 6D,F,H) and was consistently seen in three independent experiments (Figure 6E,G,I). To investigate the spatial distribution of homed T cells, we performed immunofluorescence on spleen sections of WT and CD112-deficient mice (Appendix A). When analysing spleens at 2.5 h after injection, we observed a significant reduction in adoptively transferred T cells in the T cell areas of CD112^−/−^ spleens (Appendix A). To better understand why T cell homing into LNs might be unaffected in CD112-deficient mice, we investigated the expression of CD112 and of other adhesion molecules in HEVs (Figure 6J,K). We found that CD31^+^PNAd^+^ HEVs expressed CD112 at comparable levels to CD31^+^ BECs in the spleen (Figure 6L,M). By contrast, the expression of ICAM-1, which is crucial for T cell homing to LNs [30], was approximately threefold higher in LN-derived PNAd^+^ BECs as compared to the CD31^+^ BECs isolated from spleen (Figure 6M,N). Taken together, we identified a new role for CD112 in vivo in regulating T cell entry into the spleen.

## 4. Discussion

In this study, we have investigated the role of CD112 in endothelial cell biology and in T cell trafficking in vivo. We detected CD112 expression in the murine blood vasculature and found that loss of CD112 in vivo significantly increased blood vessel coverage in retina and spleen during development and adulthood, respectively. Nectin-2 and nectin-like molecules such as CD155 have been implicated in contact inhibition, a process in which cells decrease their cellular motility and proliferation upon establishing robust cell-to-cell connections [13,31]. In a previous in vitro study with outgrowth endothelial cells [13], knockdown or blockade of CD112 enhanced endothelial cell proliferation, migration and tube formation. Similarly, we observed enhanced tube formation upon CD112 blockade in human dermal LECs. When we analysed the expression of proangiogenic factors, we detected a marginal downregulation of VEGF-A and no differences in Ang1 and Ang2 expression in CD112^−/−^ spleens. This further suggests that the enhanced coverage with blood vessels observed in the spleen and retina of CD112-deficient mice is not caused by aberrant expression of angiogenic factors but is rather linked to dysfunctional contact inhibition. It remains unclear which molecular interactions between adjacent endothelial cells is responsible for the observed effects. While human and mouse endothelial cells expressed both CD112 and CD113 in vitro, we only detected CD112 protein in blood and lymphatic vessels in mice. This suggests that—unless any other yet unknown CD112 ligand is involved—homophilic CD112-CD112 interactions likely regulate angiogenic processes in the murine vasculature.

Surprisingly, we only observed a minimal blood vascular defect in the ear skin of adult CD112^−/−^ mice, compared to the profound difference in blood vessel coverage observed in the spleen. The vasculatures of both organs are uniquely adapted to their functional role and consequently differ morphologically in many ways [28,32]. The immunological role of the spleen is to filter antigen and pathogens from the blood. Consequently, its vasculature consists of a large and highly permeable meshwork [33]. By contrast, the skin, as a barrier tissue, contains a more structured microvasculature consisting of defined and rather impermeable vessels. The latter feature tight endothelial cell–cell junctions that prevent skin-invading pathogens or larger pathogenic constituents from directly entering into systemic circulation [34]. Thus, it is likely that the dependence and role of adhesion molecules greatly varies between the two organ-specific vasculatures. In the ear skin of CD112-deficient mice, we detected a significant upregulation of VE-cadherin in LECs but not in BECs. While elevated VE-cadherin levels might therefore have masked a lymphangiogenic defect in dermal LECs, it remains possible that other molecules compensated for loss of CD112 in dermal BECs.

Previous in vitro findings already implicated CD112 in transendothelial migration of human T cells [11]. Consistent with this, we observed that CD112 blockade decreased human T cell transmigration across LEC monolayers. Furthermore, we found that antibody-based blockade of CD112 also significantly reduced transendothelial migration of murine T cells, although this effect was somewhat less striking compared to human T cell transmigration. A possible explanation could be that the anti-human CD112 antibody (clone R2.525), which we found to bind murine CD112, is less potent in functionally blocking murine as compared to human CD112. Interestingly, the absence of CD112 on LECs resulted in a profound reduction in murine T cell transmigration. In the case of human T cell transmigration, CD113 was identified as the main lymphocyte-expressed binding partner of endothelial CD112 [11]. Given the absence of both CD113 and CD112 expression in murine T cells, another CD112 binding partner must be mediating this interaction in the murine setting. Our CD112 binding partner analysis revealed that all murine T cells expressed CD226, whereas TIGIT was only expressed by a subset of effector and memory T cells. Whereas CD112 is widely expressed by antigen-presenting cells, its binding partner CD112R is an only recently identified inhibitory receptor expressed on T cells and shown to suppress T cell activation [15]. CD112R expression was detected on the majority of CD8^+^ effector/memory T cells in human blood, but blood-derived human CD4^+^ T cells were found to only express the receptor upon in vitro activation [15]. Considering that we could not detect CD112R by flow cytometry on murine CD4^+^ T cells, we suspect that it is not expressed—or at least not constitutively expressed—by the CD4^+^ T cell subsets, which mainly consist of naïve (CD44^lo^, CD62^hi^) T cells, that we analysed. The fact that the CD112 blockade did not impact transmigration of in vitro generated T_H_1 cells, in combination with the observation that T_H_1 cells—in contrast to naïve and endogenous effector/memory CD4^+^ T cells—display a striking downregulation of CD226, at present suggests that CD226 might be the T cell-expressed binding partner of endothelial CD112. However, we cannot exclude that another, so far unknown ligand on murine T cells is involved in CD112-dependent transmigration.

In mice, we found that adoptively transferred T cells homed less into the spleen in the absence of CD112, whereas T cell extravasation into LNs was unaffected. Homing into LNs has been extensively investigated and takes place in HEVs—i.e., highly specialized blood vessels optimally adapted to facilitate leukocyte extravasation [30,35]. T cell extravasation in HEVs depends on endothelial expression of the L-selectin-ligand peripheral node address in PNAd, the chemokine CCL21, and the adhesion molecule ICAM-1 [30]. By contrast, the spleen lacks such defined vessels, suggesting that extravasation relies on other molecules and mechanisms [36]. Another apparent difference is the blood flow velocity, which is much lower in the spleen as compared to the LN HEVs [36,37]. Despite similar expression of CD112, we found ICAM-1 to be significantly more highly expressed on PNAd^+^ BECs from HEVs than on splenic CD31^+^ BECs. The crucial role of LFA-1/ICAM-1 in T cell homing to LNs [30] suggests that CD112 might be redundant in HEVs due to the high ICAM-1 expression.

The involvement of LFA-1/ICAM-1 in lymphocyte entry into the spleen is still uncertain [38,39,40]. Thus far, the only other molecule reported to mediate T cell entry into the spleen is the common lymphatic endothelial and vascular endothelial receptor-1 (CLEVER-1) [41]. Interestingly, the latter study demonstrated the involvement of CLEVER-1 in lymphocyte entry to the spleen via the red pulp, thereby reporting an additional entry route for lymphocytes besides the one through the marginal zone. The fact that we detected CD112 expression in all splenic BECs could suggest its involvement in both entry processes. It is worth mentioning that the T cell homing defect in CD112-deficent mice was observed in spite of the significantly increased splenic blood vessel coverage found in these animals. This might appear somewhat counterintuitive, if one assumes that with an increased presence of blood vessels T cells should in theory have more possibilities for exiting from the vasculature into the spleen. These findings therefore suggest that T cell homing to the spleen relies more on the presence of specific adhesion molecules, such as CD112 or CLEVER-1 for adhesion and transmigration, than on the size of the vascular bed. On the other hand, it cannot be ruled out at this point that loss of CD112 might alter CLEVER-1 or ICAM-1 expression in splenic blood vessels and thereby indirectly contribute to the homing defect observed in CD112-deficient mice. Moreover, the exact exit routes of T cells out of the spleen are also not well characterized. Therefore, it remains possible that the increased blood vessel coverage observed in CD112^−/−^ spleens might additionally enhance T cell exit from this organ.

Overall, the spleen possesses characteristic immunological functions by providing protection against systemic infections. Identification of new molecules involved in splenic transendothelial migration will contribute to enhancing our understanding of this important process.

## Figures and Tables

**Figure 1 cells-10-00169-f001:**
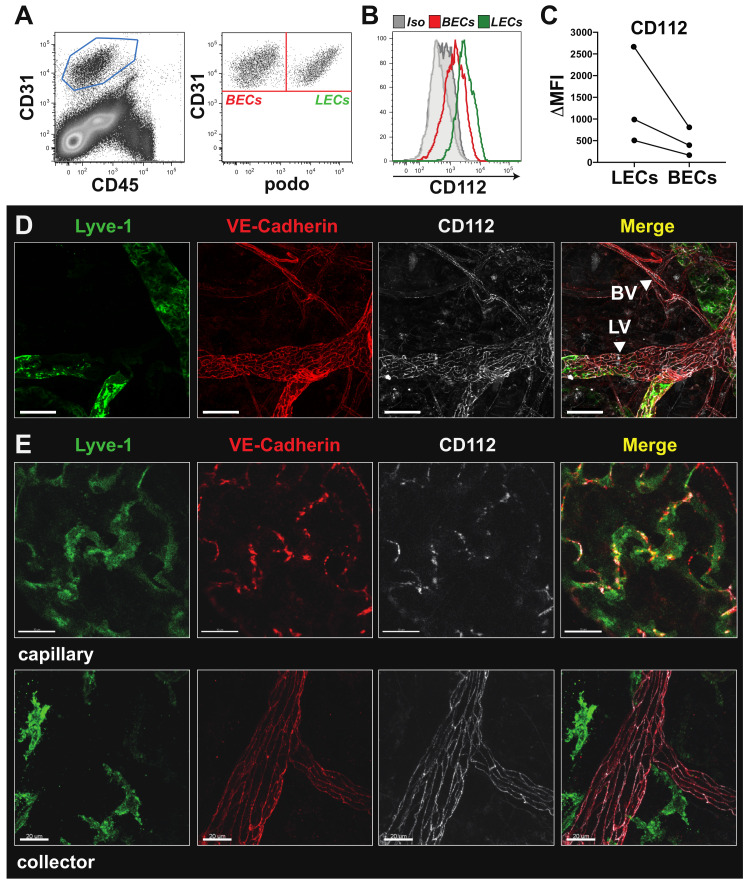
CD112 is expressed by endothelial cells in vivo. (**A**,**B**) Flow cytometry analysis of mouse ear skin single-cell suspensions. (**A**) Endothelial cells were identified as CD31^+^CD45^−^ cells (left) and further divided into blood endothelial cells (BECs) and lymphatic endothelial cells (LECs) based on podoplanin (podo) expression (right). (**B**) Representative flow cytometry plot showing in vivo expression of CD112 by LECs (green; CD31^+^podo^+^) and BECs (red; CD31^+^podo^−^). (**C**) Summary of median fluorescent intensity (MFI) values of CD112 expression on murine LECs and BECs of three experiments. Data points from the same experiment are connected by a line. (**D**) Low magnification confocal images of ear skin whole-mount immunofluorescence staining visualizing CD112 expression (white) by lymphatic vessels (LVs) and blood vessels (BVs), indicated by white arrow heads. Scale bar: 50 μm. (**E**) High magnification confocal images revealed a button-like expression pattern for CD112 (white) in lymphatic capillaries (LYVE-1^+^/VE-cadherin^+^ upper panel), respectively, and a zipper-like expression pattern in lymphatic collectors (LYVE-1^−^/VE-cadherin^+^ lower panel). Scale bar: 20 μm.

**Figure 2 cells-10-00169-f002:**
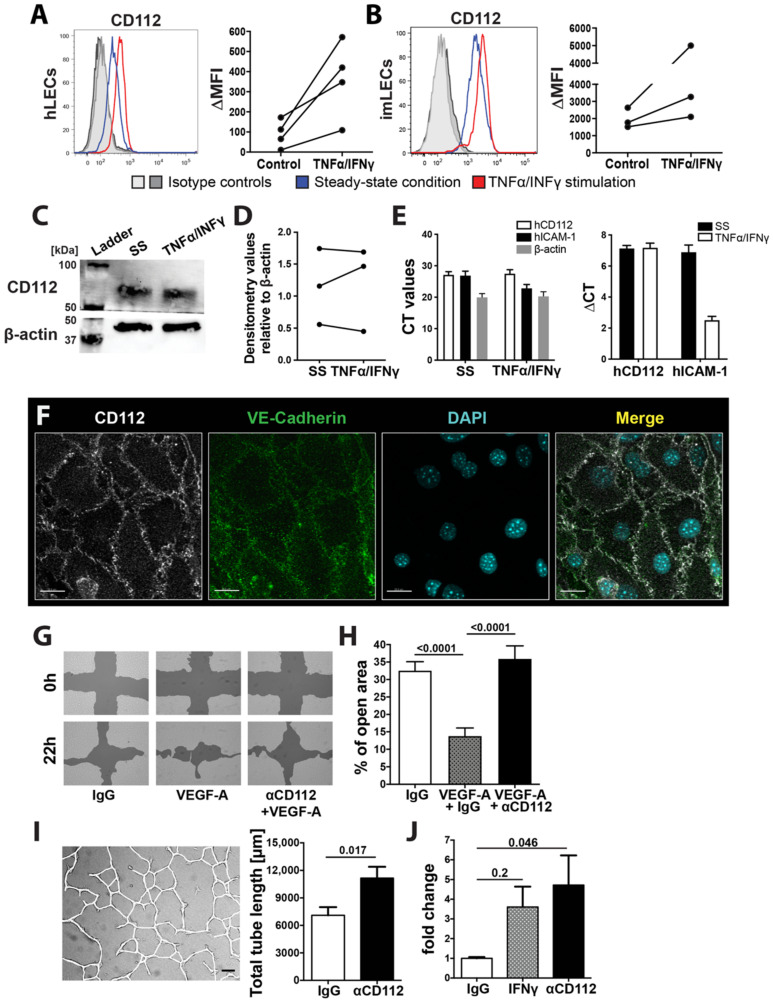
CD112 regulates endothelial cell function in vitro. (**A**,**B**) Flow cytometry analysis showing CD112 expression by human (**A**) and conditionally immortalized murine LECs (imLECs) (**B**). Representative flow cytometry plots of CD112 expression comparing steady-state (blue line) and inflamed conditions (red line: TNFα/IFNγ treated; grey lines: isotype controls) are shown on the left. Summary of MFI values of CD112 expression of three to four experiments are shown on the right. Data points of the same experiment are connected by a line. (**C**) Western blot analysis configure 112. protein (~65 kDa) expression in in vitro cultured human LECs in steady-state (SS) and upon TNFα/IFNγ-mediated inflammation. β-actin was used as an internal control. Blots were cut at the 50 kDa ladder mark. Representative data from one out of three experiments are shown. (**D**) Summary of densitometry signal intensities of CD112 protein in steady state (SS) and upon TNFα/IFNγ-mediated inflammation normalized to β-actin signal of three Western blot experiments. Data points from the same experiment are connected by a line. (**E**) Quantitative real-time PCR analysis of CD112 mRNA levels in in vitro cultured human LECs in steady-state (SS) and upon TNFα/IFNγ-mediated inflammation. Induction of ICAM-1 mRNA levels by TNFα/IFNγ was analysed as a positive control. Delta cycle threshold (CT)values: Difference between the CT values measured for CD112 and for the housekeeping gene RPLP0. Pooled data (experimental means) from three independent experiments (biological replicates) are shown. Each experiment was performed in triplicate (technical replicates). (**F**) Representative immunofluorescence images of CD112 expression (white) on in vitro cultured imLEC monolayers colocalizing with the junctional molecule VE-cadherin (green) at the intercellular junctions. Scale bar: 20 μm. (**G**,**H**) Human LEC migration upon CD112 blockade was investigated in an in vitro scratch assay. (**G**) Representative images of an LEC scratch assay in VEGF-A-induced wound closure. Scale bars: 200 µm. (**H**) Pooled quantitative analysis from three independent experiments are shown. (**I**) Representative image (left) of an in vitro tube formation assay. Confluent human LEC monolayers were incubated overnight in collagen gel solution supplemented with blocking antibody for CD112 or isotype control. Total tube length was manually analysed using a self-made macro in FIJI (ImageJ). Quantitative analysis (right) of total tube length upon CD112 blockade. One out of three similar independent experiment is shown. (**J**) In vitro permeability assay, in which 70 kDa FITC dextran was applied onto confluent pretreated LEC monolayers grown on a Boyden transwell membrane. After 30 min, diffused FITC dextran was quantified in the lower chamber. Pooled data from three similar and independent experiments are shown.

**Figure 3 cells-10-00169-f003:**
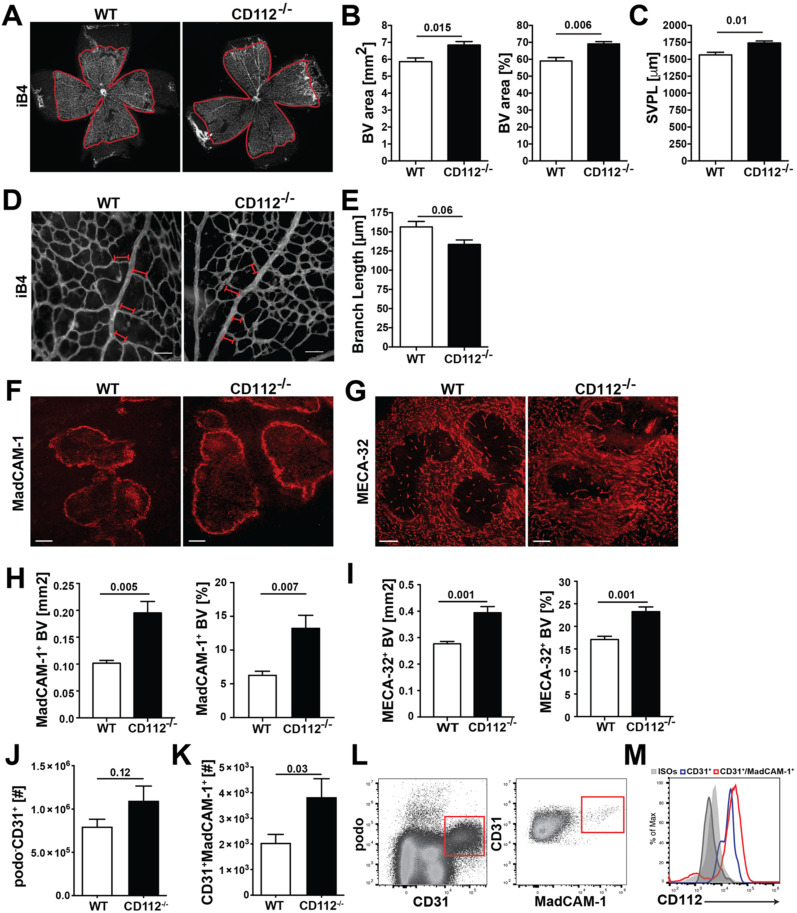
CD112 deficiency increases blood vessel coverage in the retina and spleen in mice. (**A**–**E**) Whole mounts from postnatal 6 (P6) retinas were stained with isolectin B4 (iB4) to visualize the blood vessels of the superficial vascular plexus, followed by quantification of vascular parameters. (**A**) Representative confocal images of iB4 stained P6 retinal whole mounts of CD112^−/−^ and Wild type (WT) pups. 4× objective 1.0 zoom. Image-based morphometric analysis showing (**B**) the iB4+ area (left) and percent covered by blood vessels (right) and (**C**) the superficial vascular plexus length (SVPL). Scale bar: 500 µm. Data from one out of 3 similar experiments (*n* = 4–6 mice/group). (**D**,**E**) Upon vessel maturation, vessels undergo pruning, which can be quantified by analysing the blood vessel density. (**D**) Representative images of P6 retinas of WT and CD112^−/−^ stained with iB4. 20× objective, 1.0 zoom, scale bar: 80 µm. Analysis of (**E**) the branch length formed in the peri-arterial space in CD112^−/−^ mice. Data from one experiment are shown (*n*= 4–6 mice/group). (**F**–**M**) Immunofluorescence and flow cytometry analysis of spleen single-cell suspension. (**F**–**I**) Sections sizes of 40 μm of optimum cutting temperature (OCT)-frozen spleens from CD112^−/−^ and WT mice were stained for MadCAM-1 (**F**) and MECA-32 (**G**). Representative pictures are shown. Scale bar: 200 µm. (**H**,**I**) Quantification of the MadCAM-1^+^ (**H**) and MECA-32^+^ (**I**) area in spleen of CD112^−/−^ mice; *n* = 5 mice/group. (**J**,**K**) Quantification of the number of (**J**) podo^−^CD31^+^ and (**K**) CD31^+^MadCAM-1^+^ BECs present in the spleen of WT and CD112^−/−^ mice. Pooled data from three experiments are shown. (**L**) Gating strategy of spleen single-cell suspensions. BECs were gated on CD45^−^podo^−^CD31^+^ (left), MadCAM-1^+^ CD31^+^ (right). (**M**) Representative histogram of several independent experiments showing CD112 expression by BECs in the spleen. Blue: CD31^+^, red: CD31^+^MadCAM-1^+^, grey: isotype controls.

**Figure 4 cells-10-00169-f004:**
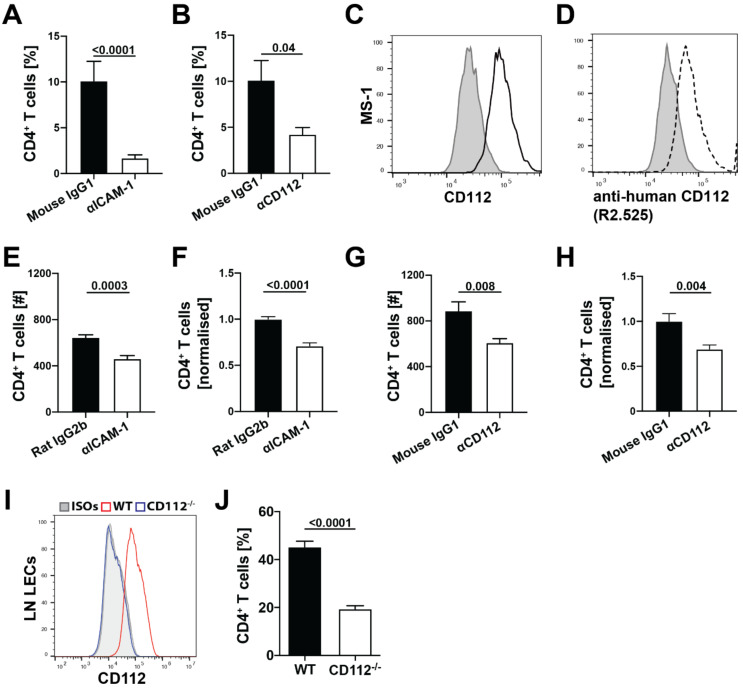
CD112 regulates CD4^+^ T cell transmigration through BECs in vitro. (**A**,**B**) In vitro transmigration assay with human T cells and primary human LECs. Quantification of transmigration efficiency upon either ICAM-1 antibody blockade (clone BBIG-I1, mouse IgG1) (**A**) or CD112 antibody blockade (clone R2.525, mouse IgG1) (**B**). Pooled data from three similar experiments are shown (*n* = 3 replicates per group and per experiment, i.e., 9 per pooled group). (**C**) Flow cytometry analysis showing CD112 protein expression by cultured blood vascular MS-1 cells. Representative histograms from three independent experiments are shown (black: CD112, grey: isotype control). (**D**) Representative histograms demonstrating the cross-reactivity of mouse anti-human CD112 antibody (R2.525, R&D systems) with mouse CD112 (black doted: antibody binding to CD112; grey: isotype controls). (**E**–**H**) In vitro transmigration assay of freshly isolated murine CD4^+^ T cells across blood vascular MS-1 monolayers in presence of anti-ICAM-1 (clone YN1, rat IgG2b) (**E**,**F**) or anti-CD112 (clone R2.525, mouse IgG1) (**G**,**H**). The number of transmigrated T cells was quantified by flow cytometry after 4 h. (**E**,**G**) show absolute numbers of transmigrated cells and (**F**,**H**) show values normalized to the isotype control. Pooled data from three similar experiments are shown (*n* = 3 replicates per group and per experiment, i.e., 9 per pooled group). (**I**) Representative histogram showing CD112 expression in WT (red) and CD112^−/−^ (blue) primary LN LECs compared to isotype controls (grey). Data from one out of three independent experiment are shown. (**J**) In vitro transmigration of freshly isolated murine CD4^+^ T cells across primary WT or CD112^−/−^ LN LECs. Pooled data from four similar experiments are shown (*n* = 4 replicates per group and per experiment, i.e., 12 per pooled group).

**Figure 5 cells-10-00169-f005:**
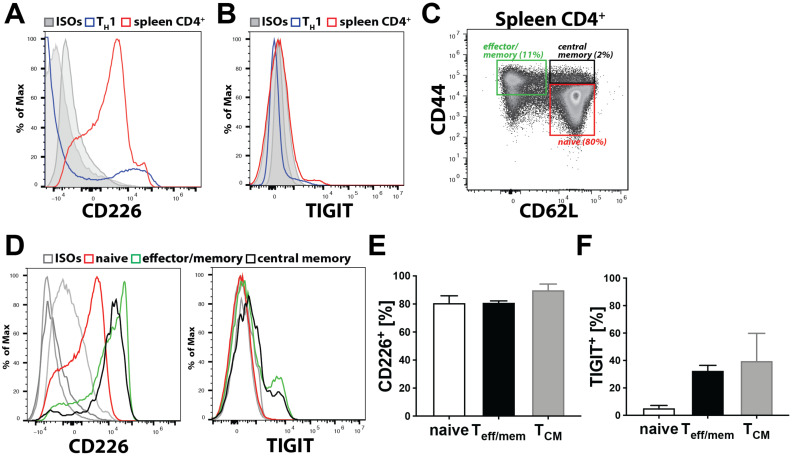
CD4^+^ T cells express CD226. (**A**–**F**) Analysis of CD112 binding partner expression on T cells. CD226 (**A**) and T cell immunoreceptor Ig tyrosine (TIGIT) (**B**) expressions on freshly isolated splenic CD4^+^ T cells (red) and in vitro generated T_H_1 cells (blue). (**C**) Characterisation of naïve, effector/memory and central memory T cells based on CD44 and CD62L expressions of splenic CD4^+^ T cells. (**D**) CD226 (left) and TIGIT (right) expression on naïve (red), effector/memory (green) and central memory (black) T cells. Representative graphs from one out of three similar experiments are shown. (**E**,**F**) Percentages of CD226^+^ (**E**) or TIGIT^+^ (**F**) T cells. T_eff/mem_: effector/memory; *Table 3*. 5. CD112 Deficiency Reduces Homing of T Cells to the Spleen.

**Figure 6 cells-10-00169-f006:**
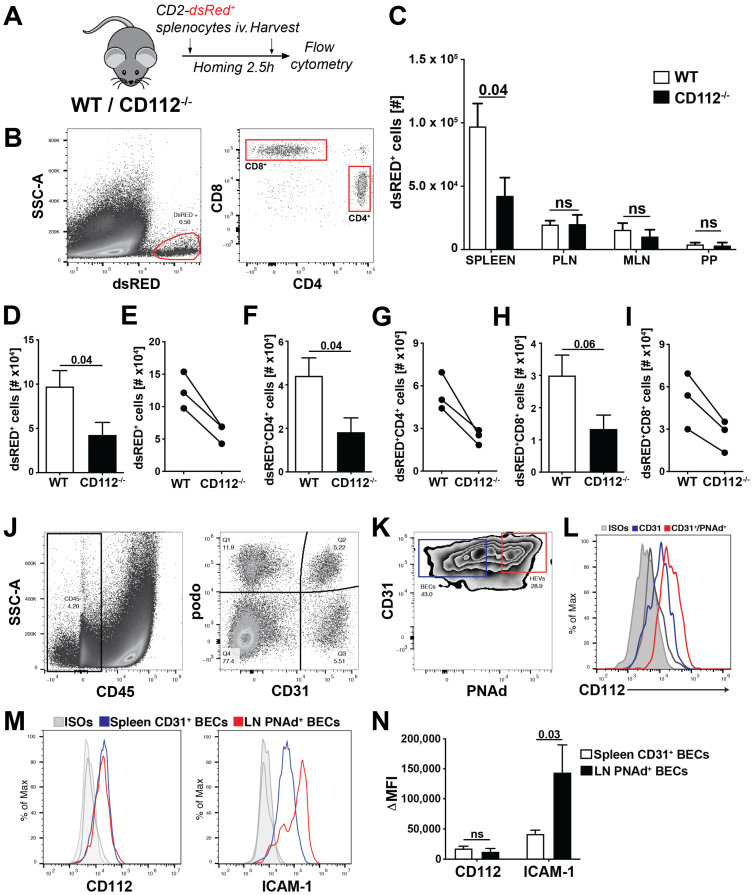
Absence of CD112 reduces homing of adoptively transferred T cells to spleen. (**A**–**I**) T cell homing experiment into secondary lymphoid organs (SLOs). (**A**) CD2-dsRED^+^ leukocytes were injected intravenously into either WT or CD112-deficient mice. After 2.5 h, SLOs were harvested and the number of homed T cells was quantified with flow cytometry. (**B**) Representative gating strategy of adoptively transferred CD2-dsRED^+^ T cells in murine spleen. Single-cell suspensions were gated on CD45^+^CD2-dsRED^+^ and further subdivided into CD4^+^/CD8^+^ cells. (**C**) Numbers of adoptively transferred CD2-dsRED^+^ T cells in different SLOs (PPs: Payer’s patches, MLN: mesentery LN, PLN: peripheral LN). Representative data from one out of three experiments are shown. *n* = 5–7 mice/group, ns = not-significant. (**D**–**F**) Quantification of (**D**,**E**) total CD2-dsRED^+^, (**F**,**G**) CD2-dsRED^+^CD4^+^ and (**H**,**I**) CD2-dsRED^+^CD8^+^ T cells in the spleen of WT and CD112^−/−^ mice from. (**D**,**F**,**H**) show representative data from one out of three independent experiments (*n* = 5–7 mice/group/experiment) performed. (**E**,**G**,**I**) show a summary of the mean values obtained in the three individual experiments. Data points (WT—CD112^−/−^) from the same experiment are connected by a line. (**J**–**L**) Analysis of CD112 expression in LN HEVs. (**G**) Gating strategy of LN single-cell suspensions. CD45^−^ stromal cells were further divided by CD31 and podo expression. (**K**) HEV BECs were identified as podo^−^CD31^+^PNAd^+^ cells. Representative plots from three independent experiments are shown. (**L**) Representative histogram showing CD112 expression in CD31^+^ LN BECs (blue) and CD31^+^PNAd^+^ HEV BECs (red). (**M**,**N**) Comparison of CD112 and ICAM-1 expression in CD31^+^PNAd^+^ LN BECs and splenic CD31^+^ BECs. (**M**) Representative histograms and (**N**) quantification of the median fluorescent intensities (MFIs) from three independent experiments.

## Data Availability

The data presented in this study are openly available in the ETH Research Collection at [doi: 10.3929/ethz-b-000462615].

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
