# Peer review of "CD112 Regulates Angiogenesis and T Cell Entry into the Spleen"

_cells, 2021, doi:10.3390/cells10010169_

Round 1

Reviewer 1 Report

In this manuscript, E. Russo and collaborators report the characterization of vascular alterations in the spleen of CD112-deficient mice as well as T cell homing defects to the spleen.  Overall, this is a well done study in which the conclusions are supported by the results. High quality images of the vascular and/or lymphatic organization of the skin, postnatal retinas and spleens show specific vascular alterations in the spleen and retinas of CD112-/- animals. Increased number of endothelial cells and blood vessels are found in the spleen of knock-out mice, prompting the authors to explore lymphocyte transmigration across CD112-/- endothelial cells and homing to the spleen of knock-out mice. Results that are shown support a function for CD112 in lymphocyte transmigration and homing to the spleen, however the interplay between the increased vascular coverage observed in CD112-/- mice and the defective lymphocyte homing is not directly addressed. I have only minor points.

Minor points:

  1. Fig 4E and F: In the text it is written that the anti-CD112 mAb reduced transmigration in a comparable manner to anti-ICAM-1 blockade. However, the figure shows the absolute numbers of transmigrated cells which are already altered by isotype control antibodies: the level of TEM with Rat IgG2b isotype is lower than mouse IgG1 isotype and is similar to anti-CD112 blockade. Therefore, adding a panel showing the percentage of inhibition as compared to the respective isotype controls will help the reader understanding the text (without removing actual panels 4E and F which are appropriate in showing the variability of TEM experiments).
  2. Fig 6 and discussion: it would be interesting to explore ICAM-1 and CLEVER-1 expression by immuno-histochemistry and quantitative RT-PCR in order to better argue about the possible explanation of the discrepancy between decreased homing to the spleen and increased splenic blood coverage in CD112-/- mice.

Author Response

Please see the attachment. Many thanks.

Reviewer 2 Report

The manuscript by Russo et al investigated the function of CD112 (nectin-2) in endothelial cell biology and leukocyte trafficking in vitro and in vivo. In vivo they show that CD112-/- mice have expanded vascular beds in the retina and spleen. Antibody blocking experiments show that anti-CD112 reduced VEGF induced cell EC migration but enhances EC tube formation. In vivo, T cell homing to the spleen was reduced in CD112-/- mice. 

The manuscript is very well written, and the materials and methods detailed enough for others to reproduce the work. The experiments are well designed with all of the necessary controls. The data are well presented, and support the conclusions well. The novel findings of this work are that CD112 plays a role in trafficking of T cells in and out of the spleen. 

Author Response

No replies to Reviewer 2 needed.